

# Classification of melanonychia, Beau's lines, and nail clubbing based on nail images and transfer learning techniques

Derya Yeliz Coşar Soğukkuyu[1] and Oğuz Ata[2]

[1] Institute of Graduate Studies, Altinbas University, İstanbul, Turkey
[2] Department of Information Technology, Altinbas University, İstanbul, Turkey

## ABSTRACT

**Background:** Nail diseases are malformations that appear on the nail plate and are classified according to their own signs and symptoms that may be related to other medical conditions. Although most nail diseases have distinct symptoms, making a differential diagnosis of nail problems can be challenging for medical experts.

**Method:** One early diagnosis method for any dermatological disease is designing an image analysis system based on artificial intelligence (AI) techniques. This article implemented a novel model using a publicly available nail disease dataset to determine the occurrence of three common types of nail diseases. Two classification models based on transfer learning using visual geometry group (VGGNet) were utilized to detect and classify nail diseases from images.

**Result and Finding:** The experimental design results showed good accuracy: VGG16 had a score of 94% accuracy and VGG19 had a 93% accuracy rate. These findings suggest that computer-aided diagnostic systems based on transfer learning can be used to identify multiple-lesion nail diseases.

## INTRODUCTION

According to World Health Organization (WHO), nearly 900 million people throughout the world are affected by skin disorders, with the most prevalent being onychosis, acne, atopic dermatitis, eczema, psoriasis, and rosacea (*Hay et al., 2014*). Although some of these diseases have similar symptoms, such as dry skin, open sores, lesions, ulcers, peeling skin, rashes, or red skin, they are generally diagnosed by healthcare providers who visually examine the skin or apply dermoscopic tests (*Jean, 2013*). Diagnosing the common nail diseases Beau's lines, melanonychia (also called blackline), and nail clubbing can be challenging, takes time, and can depend on the clinician's skills (*Tully, Trayes & Studdiford, 2012*).

Beau's lines are horizontal depressions that rise from the nail's base and spread outward from the white, moon-shaped section of the nail bed. The width of the lines can be used to determine how long the disease has been present (*Braswell, Daniel & Brodell, 2015*).

Melanonychia, also known as blackline, is a discoloration that causes a black or brown line that spreads longitudinally across the nail's entire length. The discoloration can sometimes be scattered or transverse (*Tosti, Piraccini & de Farias, 2009*).

Corresponding author
Derya Yeliz Coşar Soğukkuyu,
yelizsogukkuyu@gmail.com

Nail clubbing is when the nail appears broader, sponge-like, or like a bloated spoon, and is generally a sign of lung cancer (*Currie & Gallagher, 1988*).

Approximately 10% of all dermatologic diseases are nail disorders and mainly occur in older people (*Tosti, 2018*). The development of prediagnosis systems has provided noticeable results both in human survival and quality of life (*Dhanvijay & Patil, 2019*). Additionally, physicians have stated that the majority of skin disease issues can be treated with medication when the symptoms have been correctly identified. A recent hot topic rising from artificial intelligence (AI) is the use of an autonomous disease diagnosis system by dermatologists to reduce their labor. Computer-assisted diagnosis also provides inexperienced clinicians the ability to use dermoscopic processes to input disease symptoms when screening difficult cases (*Wollina et al., 2016*).

AI is the science of making meaningful data from various types of datasets by simulating human intelligence processes using machines, particularly computer systems (*McCarthy, 2004*). Many studies have focused on how AI can be used to enhance or improve skin disease screening methods. Research has shown that AI has caused a fundamental change in clinical medicine (*Briganti & Le Moine, 2020*). Additionally, several studies have proposed AI models with the same performance and accuracy in classifying skin images as dermatologists (*Hamet & Tremblay, 2017*). In other words, AI's potential for improving dermatological treatment is becoming generally accepted among dermatologists (*Polesie et al., 2020*). AI has the ability to learn from images and can aid in the early diagnosis of nail diseases by gathering and analyzing relevant information in a few minutes in order to produce outcomes. Moreover, AI techniques find the patterns in patient data to enhance and optimize trial design. Although dermatologists are skilled at identifying skin diseases, additional cutting-edge imaging solutions like complete body scanners can offer more precise information for medical professionals. Various image processing techniques, including image acquisition, image pre-processing, image segmentation, and feature extraction, are performed for nail or skin-related computer vision tasks as non-invasive procedures (*Juna & Dhananjayan, 2019*).

A considerable number of studies have been published on skin cancer that extracted features from images and then fed them into melanoma classification models. In a study conducted in 2016, researchers trained the first neural network for melanoma detection, resulting in sensitivity and specificity values of 0.81 and 0.80, respectively (*Nasr-Esfahani et al., 2016*). At Stanford University, a study was implemented over a year on the classification of skin lesions using a GoogLeNet Inception v3 convolutional neural network (CNN) consisting of 129,450 clinical images belong to 2,032 different diseases. The results showed that for binary classification, the proposed CNN model achieved 72.1 ± 0.9% (mean ± S.D.) overall accuracy, whereas two dermatologists attained 65.56% and 66.0% accuracy (*Esteva et al., 2017*).

A detailed examination of computer vision tasks by researchers showed that deep learning approaches, which automatically learn characteristics from massive datasets, performed better than humans at skin disease classification (*Whited & Grichnik, 1998*; *Ouyang et al., 2015*; *Madooei & Drew, 2016*; *Sermanet et al., 2013*).

It has been conclusively shown that transfer learning is commonly used in skin disease classification, more specifically for nail diseases since there are limited available data. A pre-trained classification system was proposed using ResNet50 to categorize skin lesions as melanoma or nevi. The sensitivity and specificity for the suggested model were 77.9% and 82.3%, respectively (*Brinker et al., 2019*). Furthermore, a system for the multi-class classification of skin lesions, such as melanoma, seborrheic keratosis, and nevi, was proposed based on two different transfer learning algorithms, ResNet-152 and visual geometry group (VGG)-19. Two different datasets, A1 ($N = 49,567$) and A2 ($N = 3,741$), were used. The best result was taken with A1 sensitivity as well as specificity area under the curve with values of 96.0% and 94.7%, respectively (*Han et al., 2018*).

Additionally, specific cardiovascular diseases based on the color of the fingernails were identified in order to propose a classification system using ANN (*Tolentino et al., 2018*). The ANN disease classification model was proposed by researchers for eczema with an accuracy of 63.01% (*De Guzman et al., 2015*). During AI training, the system required large clinical image datasets. Therefore, the researchers created their own research datasets of nail pictures to diagnose onychomycosis, and developed a model using the pre-trained ResNet-152 and VGG-19 networks. For the final prediction, the outputs of the two networks were merged and fed into a two-layered feedforward neural network. According to experimental findings, deep learning showed a better ability to diagnose onychomycosis than most of the dermatologists who participated in the study. Results showed that the test sensitivity, specificity, and area under the curve values were 96.0%, 94.7%, and 0.98%, respectively (*Han et al., 2018*).

*Nijhawan et al. (2017)* developed a novel model to identify 11 types of nail diseases (onychomycosis, subungual hematoma, Beau's lines, yellow nail syndrome, psoriasis, hyperpigmentation, koilonychias, paronychia, pincer nails, leukonychia, and onychorrhexis) using a hybrid of CNNs with an accuracy of 84.58% (*Nijhawan et al., 2017*). This view was supported by *Decroos et al. (2021)* who argued that deep learning algorithms are viable when diagnosing nail diseases and achieved 0.981 accuracy in his research.

Results data from several studies showed that using deep learning techniques reduced human error in the field of image processing. The process of deep learning involves training a model on a vast amount of data, and the model's weight and bias are learned simultaneously in order to use it in other models as a pre-trained model. There are many pre-trained architectures available for researchers to speed up the learning process and minimize computational power on large datasets (*Krishna & Kalluri, 2019*).

To address this issue and implement an autonomous system, an efficient AI-based model that classified nail diseases using image data was implemented. This study also aimed to reduce human diagnostic errors by adapting AI techniques. Three common nail disease types, Beau's lines, melanonychia, and nail clubbing, were chosen for classification using VGGNet architecture with clinical image data. This study proposes a clinical decision system that can be used by healthcare domain experts in order to provide the following significant contributions:

1) The most popular deep learning algorithm, CNN, was applied to create a novel nail disease classification model using transfer learning;

2) A medical decision-making system that provides dermatologists and healthcare authorities with the information they need to make decisions quickly and effectively with a 94% accuracy rate;

3) Modern, state-of-the-art models were used to compare the performance of the proposed ensemble model; and

4) The proposed ensemble framework achieved high prediction accuracy and therefore may be implemented in smart devices for early Beau's lines, melanonychia, and nail clubbing patient screening.

The organization of this article is structured in the following order: Materials and Methods describe the data, Methodology, and proposed framework with general concepts for our image analysis system. The experimental findings and analysis are then presented in the Results section. The article ends with Discussion and Conclusion sections.

## MATERIALS AND METHODS

This study builds on transfer learning algorithms and proposes an automated pre-diagnosis system for three types of nail diseases: Beau's lines, melanonychia, and nail clubbing. A classic deep learning process was applied according to the following procedure: selection of the network architecture according to the task to be performed, training and testing several models to optimize the parameters, and testing each model on different data to evaluate the actual performance. Extraction of skin lesions from images occurred during the segmentation stage after preventing other non-lesion structures from interfering with the diagnostic process.

In this study, VGG16 and VGG19 models were implemented because VGGNet deep learning architecture is capable of classifying 1,000 images from 1,000 different categories with 92.7% accuracy (*Kaur & Gandhi, 2019*). This methodology has a number of advantages, such as highly accurate picture categorization results, development by a prestigious and trusted team, being applicable to different cases, and free cost to the public. However, the research does not consider ResNet or DenseNet, even though they perform better than VGG in some imaging tasks. A considerable amount of literature has been published on medical datasets that primarily used VGGNet (*Suganyadevi, Seethalakshmi & Balasamy, 2022*).

### Nail disease dataset

Publicly available data on the three types of nail disease were gathered from Kaggle.

A total of 723 images were provided and no information about the patient population or any demographic details were supplied. This model was trained on 468 images and tested on 255 images where both samples had nail diseases. The categories included three different classes: Beau's line photos—Beau's lines, blackline photos-melanonychia, and clubbing disease photos—nail clubbing. The images belonging to different categories were selected arbitrarily from the whole set. All images were non-standardized, and the sizes of

**Table 1 Summary of the nail disease dataset.**

| Class labels | Training | Testing |
|---|---|---|
| BeausLine | 185 | 96 |
| Melanonychia-BlackLine | 160 | 85 |
| Clubbing | 123 | 74 |
| Total | 468 | 255 |

the images varied, so images were formatted to $150 \times 150$. The proportion of each observed category was as follows: Beau's lines 39.5%, melanonychia 34.2%, and nail clubbing 26.3%. Table 1 shows the disease distribution of the image data.

## Proposed framework

This section describes the formalization and theoretical justification of the problem. The prediagnosis of nail diseases is challenging because disease types are categorized by labels. However, images are categorized according to a range. In contrast to other deep learning techniques, transfer learning transfers knowledge from one model to another and can be successfully implemented on small datasets (*Liu et al., 2017*). Transfer learning is a useful, proven, and automatic learning technique used in computer vision by leveraging knowledge from existing models and utilizing automatic feature engineering capabilities (*Hinton, Osindero & Teh, 2006*). A deep learning model proceeds to learn low-level features in the first layers of the network, such as identifying edges, colors, and fluctuations in intensities. These features may not seem relevant to a given dataset or task. With transfer learning, pre-trained models allow the system to use the patterns that have been learned rather than starting from the beginning. For computer vision projects, one of the major challenges is collecting large data to train and test (*Gopalakrishnan et al., 2017*). In this study, the proposed framework used transfer learning based on pre-trained "VGG16" and "VGG19" models for nail disease classification due to the limited dataset. The flow chart of the proposed technique is presented in Fig. 1. For preprocessing data, the technique used in this study rescaled the images with respect to model expectations (*Taylor & Nitschke, 2018*).

## Transfer learning

Transfer learning is a method that transfers the information gained from a previous task to the target task with significantly higher performance (Fig. 2). This is achieved by adapting a pre-trained model, which is selected as a base for the problem-specific requirements. Freezing of layers is one of the main concepts discussed when studying transfer learning. When a layer is no longer functional for training, whether it be a CNN layer, hidden layer, or block of layers, it is expected that it be locked. As a result, during training, the weights of frozen layers are not updated whereas layers that have not been frozen continue on to the standard training process (*Brodzicki et al., 2020*).

The steps of the transfer learning process used in this study were implemented according to the procedure outlined in Fig. 3 (*Ray, 2018*).
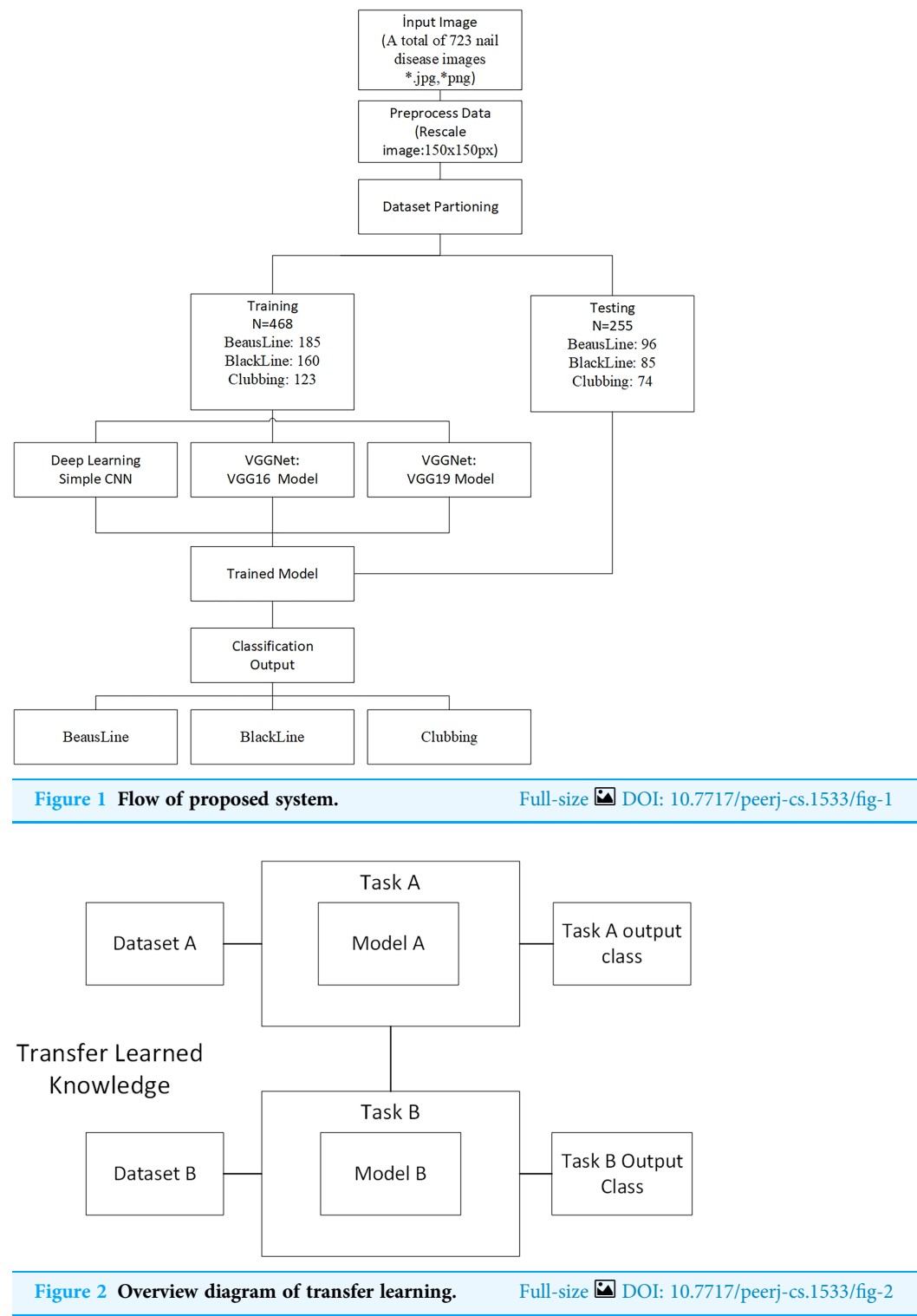

**Figure 1** Flow of proposed system.               

**Figure 2** Overview diagram of transfer learning.   

Step 1: Selection of pre-trained model. A pre-trained model that is most applicable to the problem is chosen as the base of the training phase. For computer vision tasks, AlexNet, VGGNet, ResNet, Inception, and DenseNet models are advised.

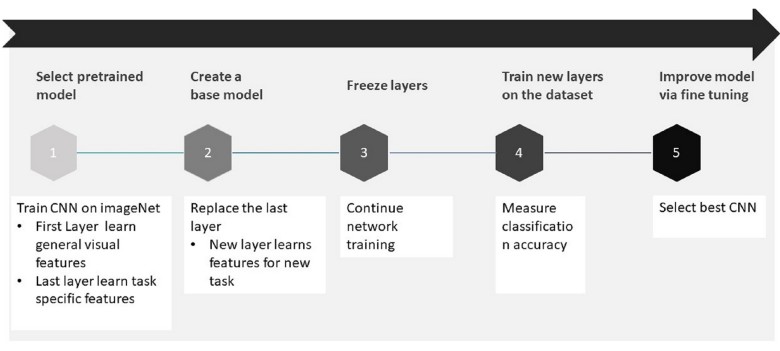

**Figure 3** **Transfer learning process.**

Step 2: Creation of a base model. The base model is one of the models that is chosen in the first phase to be closely our objective, such as VGG16 or ResNet-50. Network weights are downloaded to avoid training the model again.

Step 3: Freezing the layers. After choosing the base model in step 2, the next step is to freeze the initial layers to prevent the models from relearning the fundamental features. If layers are not frozen, all of the learned knowledge can be lost and time wasting can occur during the training.

Step 4: Adding new trainable layers. The feature extraction process is conducted from the layers through the basic model. New trainable layers are added and trained on the model to predict the specific task outcomes since the final output of base model will be different according to the desired task.

Step 5: Fine-tune the model. Up until this stage, the model has been able to make predictions, and less attention has been given to performance. To increase the performance of the model, fine-tuning is applied by unfreezing the base model or a part of it and retraining the entire model on the full dataset at a very low learning rate.

## Pre-trained models

Pre-trained models were trained on large datasets and solved similar, common task problems by including neural network biases and weights for the features of the dataset they were trained on. In this study, VGG16 and VGG19 models were used as base models for feature extraction.

## Transfer learning algorithm selection

VGG, also known as VGGNet, is a classical CNN architecture and is considered one of the best computer vision models. It is a standard deep CNN architecture with multiple layers: an input layer, an output layer, and various hidden layers. VGG was developed to significantly improve model performance by increasing the depth using an architecture with very small (3 × 3) convolution filters.

## VGG16

VGG-16 architecture consists of 16 convolutional layers, three of which are fully connected with 138 trainable parameters (*Zeinali et al., 2019*), as described in Fig. 4.

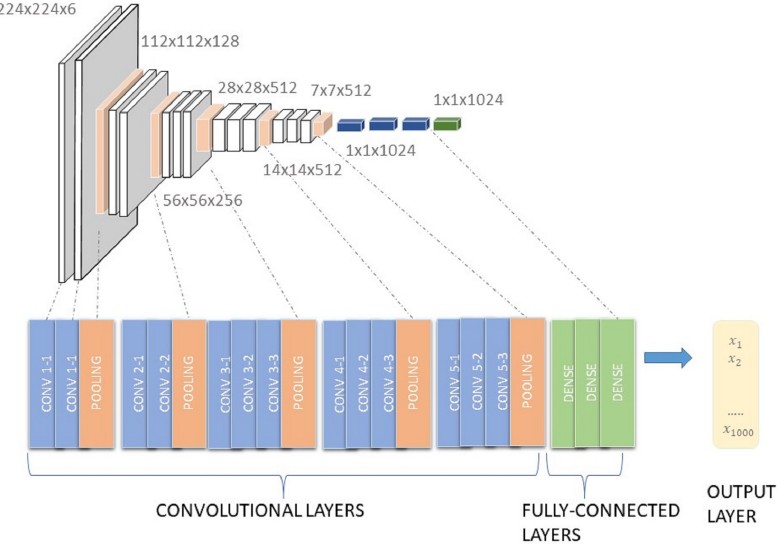

**Figure 4 Architecture of VGG16.**

The VGG model that supports 16 layers is also referred to as VGG16 and is used for object detection and a classification CNN model proposed by *Simonyan & Zisserman (2014)*. The VGG16 model achieved 92.7% test accuracy and was able to classify 1,000 images from 1,000 different categories in ImageNet. ImageNet is a dataset consisting of more than 14 million images belonging to nearly 1,000 classes. It replaced the large kernel-sized filters with several simultaneously-placed 3 × 3 kernel-sized filters (*Qassim, Verma & Feinzimer, 2018*).

## VGG19

A CNN with 19 layers (VGG-19) used the ImageNet database in a pre-trained version of the network with more than a million images. The pre-trained network had the ability to categorize 1,000 different photos of objects, including a keyboard, a mouse, and a pencil. The model acquired rich feature representations for a variety of images and accepted images with a resolution of 224 × 224 (Fig. 5).

## Performance measures

Increasing prediction accuracy is the main objective of model training because it shows how well the model performs a given task. In this study, generally acceptable metrics were used for calculating the model's performance (Table 2).

The metrics "R squared," "mean absolute error (MAE),", "mean squared error (MSE)," "root mean square error (RMSE)," and "accuracy" were used to calculate model performance with the following criteria (*Chicco, Warrens & Jurman, 2021*):

Results that were correctly categorized into the positive class represented true positive (Tp) predictions, results that were correctly categorized into the negative class represented true negative (Tn) predictions, results that were falsely categorized into the positive class represented false positive (Fp) predictions, and results that were falsely categorized into the negative class represented false positive (Fn) predictions.
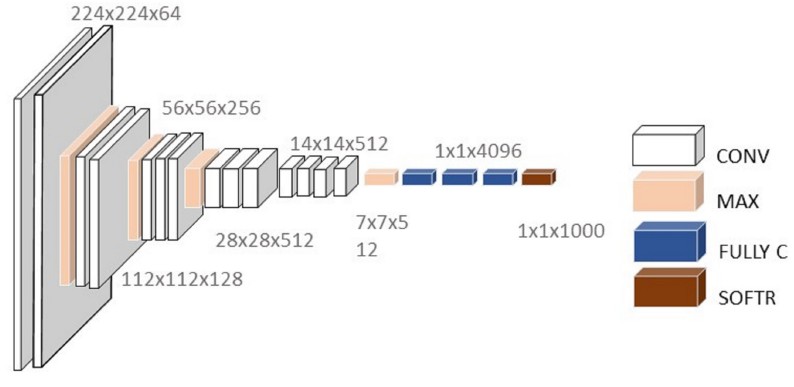

**Figure 5 Architecture of VGG19.**

**Table 2 Overview of performance measures.**

| Measure | Tabulation | Description |
|---|---|---|
| Accuracy | $\dfrac{Tp + Tn}{Tp + Fp + Tn + Fn}$ | Accurately predicted observations as a percentage of all observations |
| MAE | $MAE = \dfrac{1}{n}\sum_{i=1}^{n}\lvert y_i - \hat{y}_i\rvert$ | Mean of the absolute values of each prediction error on all instances of the test dataset. |
| MSE | $MSE = \dfrac{1}{N}\sum_{i=1}^{N}\left(y_i - \hat{y}_i\right)^2$ | The difference between your model's predictions and the ground truth, square it, and average it out across the whole dataset |
| RMSE | $RMSE = \sqrt{\dfrac{\sum_{i=1}^{N}\lVert y(i) - \hat{y}(i)\rVert^2}{N}}$ | Shows how far predictions fall from measured true values using Euclidean distance |
| R squared | $R^2 = \dfrac{\sum\left(\hat{y}_i - \bar{y}\right)^2}{\sum\left(y_i - \bar{y}\right)^2}$ | As a ratio of total variation of data points explained by the regression line (Sum of squared regression) |

# RESULTS

Two basic techniques were performed in the experimentation stage: one using traditional CNN and the other using transfer learning to classify three classes of nail disease images (Beau's lines, melanonychia, and nail clubbing).

## Experimental results

In the first experimental stage, CNN was used to train the network using the Tensor-Flow platform with open-source Keras packages and the Python programming language. The basic CNN model in experiments (Beau's lines, melanonychia, and nail clubbing) obtained 0.912 training accuracy and a 0.77 validation accuracy score with underfitting. The model was compiled regarding the following parameters:

Conv2D (32 filters of size 3 by 3), where features were extracted from the image;

MaxPooling2D, where the images were half-sized;

Flatten, where the format of the images was transformed from a 2d-array to a 1d-array of 150 150 3-pixel values;

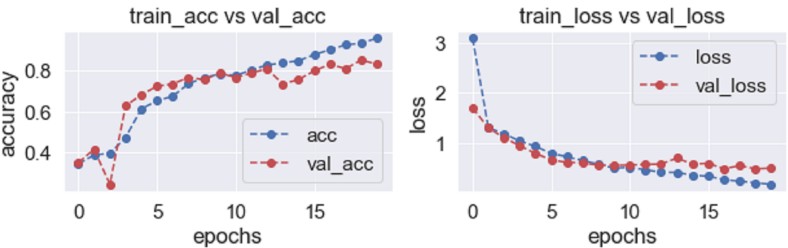

**Figure 6 Accuracy and loss graph of the proposed model basic CNN model.**

Relu, where the parameter returned max (x, 0) for given a value x;

Softmax 6 neurons, where the probability was calculated for the image belonging to a specific class;

Optimizer: Adam = RMSProp + Momentum, where Momentum considered past gradient in order to provide a better update; and

RMSProp, where the exponentially weighted average of the squares of past gradients was calculated.

The accuracy and losses in every epoch during training and validation are shown in Fig. 6.

In stage 2, VGG16 and VGG19 were implemented for transfer learning experiments. Models were trained using Adam optimizer with a binary cross-entropy loss function and a default learning rate.

Table 3 shows the obtained results of VGG16 and VGG19.

The results demonstrated that the proposed VGGNET model yielded higher classification scores on the nail disease image dataset from classical neural networks. Across the three deep learning models trained in this research, VGG16 and VGG19 considerably outperformed the remaining traditional CNN model. The model accuracy for VGG16 was 94%, and 93% for the VGG19 model, with no significant difference found.

According to the MAE, MSE, and RMSE performance measures, the VGG19 model received lower scores and therefore performed better than the VGG16 model. The results, as seen in Table 3, indicate that VGG19 had better MAE (recall), MSE, and RMSE scores (0.08, 0.133, and 0.365, respectively) than VGG16 (0.086, 0.149, and 0.386, respectively). In terms of R2 Square, VGG16 surpassed VGG19 with a score of 0.773.

A confusion matrix is one of the important ML techniques, and visually represents a model's correct and incorrect predictions.

As seen in Fig. 7, the proposed VGG16 model correctly identified 245 disease images out of 255, whereas the VGG19 model correctly identified 239 disease images out of 255.

A comparison based on the two outcomes revealed that the suggested nail disease detection system works efficiently with a transfer learning strategy.

Table 4 summarizes the related works' performance summary. In this study, we achieved better accuracy than some of the other studies in the literature and were very close to attaining the best performance score (*Goel & Nijhawan, 2019*; *Hadiyoso & Aulia, 2022*; *Regin et al., 2022*).

**Table 3 Experimental results of the study.**

| Classification results | MAE | MSE | RMSE | R2 square | Accuracy |
|---|---|---|---|---|---|
| VGG16 with fine tuning | 0.086 | 0.149 | 0.386 | 0.773 | 0.945 |
| VGG19 with fine tuning | 0.08 | 0.133 | 0.365 | 0.797 | 0.937 |

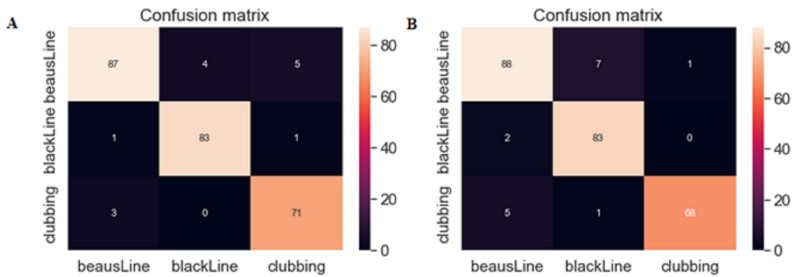

**Figure 7 (A) Confusion matrix of the VGG16 model, (B) confusion matrix of the VGG19 model.**

**Table 4 Performance comparison with related studies.**

| Reference | Classification type | Methodology | Dataset sample | Result |
|---|---|---|---|---|
| *Nijhawan et al. (2017)* | Multi-class classification: 11 nail diseases | Hybrid of convolutional neural network (CNNs) | N = 4,190 | Accuracy 84.58%. |
| *Kim et al. (2020)* | Binary classification: onychomycosis | Deep learning | N = 90 | Sensitivity 72.7%, specificity 72.9%, AUC 0.755 |
| *Regin et al. (2022)* | Binary classification: nail color change | Ensemble of CNNs | N = 185 | Accuracy of 95% |
| *Jarallah et al. (2021)* | Multi-class classification: healthy nail, nail hyperpigmentation, nail clubbing and nail fungus. | AlexNet | N = 280 | Accuracy 92.5% |
| *Jansen et al. (2022)* | Binary classification: onychomycosis | U-Net | N = 664 | Accuracy 86.49% |
| *Hadiyoso & Aulia (2022)* | Multi-class classification: koilonychia, Beau's lines, and leukonychia | Transfer learning VGG-16 | N = 333 | Accuracy 96% |
| *Goel & Nijhawan (2019)* | Binary classification: onychomycosis | Transfer learning VGG-19 | N = 100 | Accuracy 98.5% |
| *Indi & Gunge (2016)* | Binary classification: | RGB analysis | N = 100 | Accuracy 65% |
| This study | Multi-class classification: melanonychia, Beau's lines, and nail clubbing | Transfer learning VGG16 and VGG19 | N = 723 | Accuracy 94% and 93% |

# DISCUSSION

The present retrospective study is an attempt to propose a model for nail disease classification based on a non-extensive dataset of clinical images with an accuracy between 93–94%.

Outcomes in this study were consistent with the findings of previous successful studies, indicating that the proposed model has the potential to classify multiple-lesion nail diseases in the field of computer-aided diagnostics. There were similarities and less accurate results between this study and others with the highest accuracy results, 95%, 96%, and 98.5% (*Regin et al., 2022*; *Hadiyoso & Aulia, 2022*; *Goel & Nijhawan, 2019*), respectively.

*Regin et al. (2022)* achieved an AUC of 95% using VGG16, similar to our model. A possible explanation of this slight difference in results is an increase in the number of hidden layers, the five max-pooling and 16-convolutional layer.

*Hadiyoso & Aulia (2022)* achieved better performance with an AUC of 96% using VGG16, similar to our model. This could be because the parameters were changed to a batch size of 20 and different epoch variations.

It is possible to hypothesize that in an AI network, changing model parameters can result in better accuracy.

*Aishwarya, Goel & Nijhawan (2019)* achieved the best performance among all researchers with an AUC of 98.5% using hybrid ANN VGG19 that was also similar to our second transfer learning model. This data must be interpreted with caution because their dataset was limited to 100 samples for binary classification. Therefore, the performance of the model should be tested on different datasets for multi-class classification.

*Nijhawan et al. (2017)* had an accuracy of 84.58% using hybrid CNNs even though they had an adequate number of samples $N = 4,190$. This inconsistency may be due to dataset quality, the need for more data processing, and number of hidden layers used.

There are a few similar studies (using the same dataset and multi-class classification) in the literature (*Jarallah et al., 2021*; *Hadiyoso & Aulia, 2022*). In contrast to the literature, which mostly used binary classification for onychomycosis (*Kim et al., 2020*; *Jansen et al., 2022*; *Aishwarya, Goel & Nijhawan, 2019*; *Regin et al., 2022*; *Indi & Gunge, 2016*), our proposed model classified diseases across three different categories.

In this study, the results differed from other studies because we used VGG19 with a better accuracy score of 0.93, and melanonychia and nail clubbing classification were implemented using a VGG16 model for the first time.

However, the findings were subject to two limitations. The most significant was the limited open access data. Currently all available open access data is for skin disease lesions, not for nail diseases. In addition to the limited dataset, the proposed model only used images, and patient medical history was not included in the studies.

## CONCLUSIONS

The healthcare industry has seen a transformation due to the use of CNN-based deep learning methods. These algorithms are becoming more widely used since they have provided encouraging results. This research confirms previous findings and contributes to our understanding of the automatic detection of nail disease where the appropriate classification is essential. This study set out to provide a transfer learning model based on VGG16 and VGG19 for the identification of nail diseases into three classes (Beau's lines, melanonychia, and nail clubbing) using photos of nail disease disorders from Kaggle. A

total of 723 nail disease images (281 Beau's lines class images, 245 melanonychia-black line class images, and 197 nail clubbing images) were used for a training and testing process. The test was carried out with transfer learning parameters with 30 epoch variations. The results showed that the implemented model with VGG16 had a classification accuracy of 94.0% and performed well. This study refined the theory of *Nijhawan et al. (2017)*, by proofing computer aided systems that can be used for nail disease classification. This study also refined the theory of *Jarallah et al. (2021)* by proofing the usability of VGG16 and VGG19 for the multiclass classification of nail disease. This study confirmed the theory of *Hadiyoso & Aulia (2022)* by identifying melanonychia-black line nail disease with over 93.0 accuracy using VGGNet. This study confirmed the theory of *Aishwarya, Goel & Nijhawan (2019)* by proving that transfer learning can be applied successfully even if a model has a small dataset. In upcoming years, it will be possible to detect early-stage diseases by observing even slight changes in the nail, if the disease is detected early.

Further research should be conducted in order to provide more reliable results by feeding the system with electronic clinical records such as patient history, gender, and age. More research is needed to determine if there is a relationship between nail disease and skin cancer. The findings of this study have a number of important technical implications such as further augmentation of models with more comprehensive set of images, and the development of a good fine-tuning strategy where other pre-trained models, such as the DenseNet, Resnet, or Inception v3, can also be applied to improve accuracy and performance. Finally, this system can also be tested on animal nail pictures, and after training and testing in different datasets, a web page interface or a mobile application can be developed as a secondary CAD system for medical experts to use.

### Funding
The authors received no funding for this work.

### Competing Interests
The authors declare that they have no competing interests.

### Author Contributions
- Derya Yeliz Coşar Soğukkuyu conceived and designed the experiments, performed the experiments, performed the computation work, prepared figures and/or tables, and approved the final draft.
- Oğuz Ata analyzed the data, authored or reviewed drafts of the article, and approved the final draft.

### Data Availability
    Code is available in the Supplemental Files.
    Data are available at Kaggle:
    https://www.kaggle.com/datasets/arthitaya/nail-dataset.

https://www.kaggle.com/datasets/arthitaya/nail-dataset2.

https://www.kaggle.com/code/reubenindustrustech/nail-disease-image-augmentation/notebook.

## Supplemental Information

Supplemental information for this article can be found online at http://dx.doi.org/10.7717/peerj-cs.1533#supplemental-information.

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
