# Peer review of "Classification of melanonychia, Beau’s lines, and nail clubbing based on nail images and transfer learning techniques"

_PeerJ Computer Science, doi:10.7717/peerj-cs.1533_

## Round 0.1 · original submission · Major Revisions

The reviewers have substantial concerns about this manuscript. The authors should provide point-to-point responses to address all the concerns and provide a revised manuscript with the revised parts being marked in different color.

Reviewer 1 ·

Basic reporting

This manuscript describes a study that aims to develop a transfer learning model based on VGGNet, specifically VGG16 and VGG19, to classify nail diseases into three categories: Beauís Lines, Melanonychia, and Nail Clubbing. The transfer learning model was trained and tested with 30 epoch variations, resulting in a classification accuracy of 94.0% for the VGG16 model. The authors suggest that this model can support clinical diagnosis of nail diseases and enable early detection of illnesses. They also propose the possibility of using patient-reported clinical signs to feed into the disease prediction engine.

There are still some limitations:

The abstract should be rewrite. There should be include information about background, method, result and finding.

The English should be improved.

The quality of the figures needs to be improved.

Experimental design

There should include more discussion about the compare with other studies in discussion section. The limitation should be remove to the discussion part

Table 4 should include the results from this study

Figure 1 should be included more information and config detals

Validity of the findings

There are various CNN network such as VGG, ResNet, DenseNet, Inception et al and the ResNet, DenseNet performance better than VGG in some imaging tasks. Maybe author should explain why choose VGG network.

Reviewer 2 ·

Basic reporting

The manuscript by Derya Yeliz Coşar Soğukkuyu, Oğuz Ata provided a thorough background on the problem being addressed, which was the difficulty of diagnosing nail diseases and the potential for artificial intelligence (AI) to aid in this process. The authors also presented related statistics on skin disease diagnosis using AI techniques, which supports the need for further research in this area. The authors described the proposed approach in detail, including the use of transfer learning with the VGGNet architecture for creating a classification model based on public available nail disease image data. They also presented experimental findings that showed good performance of the VGG16 model with 94% accuracy for the dataset. The significance of the study's contributions was discussed in the manuscript, which included the development of an efficient AI-based recommendation system for classifying nail diseases and reducing errors by adapting AI techniques. The proposed ensemble framework achieved high prediction accuracy, making it a promising tool for early detection of Beau’s Lines, Melanonychia, and Nail Clubbing. Therefore, I recommend publishing this work in PeerJ Computer Science after addressing the following concerns:

1. The Introduction section is too long, while the Results and Conclusions sections are not enough.
2. In the Introduction section, the authors should provide more information about how AI techniques can aid in early diagnosis of skin diseases rather than just mentioning it as a solution.
3. Provide more information about the dataset used for training, such as the size and diversity of the images.
4. The authors should also discuss the limitations of the study and potential areas for future research.

Experimental design

no comment

Validity of the findings

no comment

Reviewer 3 ·

Basic reporting

no comment

Experimental design

no comment

Validity of the findings

no comment

Reviewer 4 ·

Basic reporting

The manuscript presents a novel approach to the classification of nail conditions using transfer learning techniques. The authors provide a clear and detailed description of their methodology for training a neural network to accurately classify images of nails with melanonychia, Beaus lines, and nail clubbing. They demonstrate impressive results, achieving high accuracy rates in classifying these conditions based on images.

But there still have some point to improve.

Experimental design

1. It was observed that ResNet and DenseNet outperformed VGG in some of the imaging tasks. However, the authors did not provide an adequate explanation for the choice of the VGG network in their experiments. It would be beneficial if the authors could provide more insight into their decision to choose VGG over the other CNN architectures, given that ResNet and DenseNet have shown better performance in some of the imaging tasks.

Providing a justification for the selection of the VGG network would not only enhance the clarity of the paper but also help the readers in understanding the authors' reasoning behind the choice of the CNN architecture.

Validity of the findings

It would be beneficial if the authors could elaborate on the similarities and differences between their results and those of previous studies, and how their study contributes to the advancement of knowledge in the field. This will not only improve the overall understanding of the study but also provide valuable insights for future research in the area.

Additionally, I suggest that the authors move the discussion of limitations from the methods section to the discussion section. This will enable a more comprehensive discussion of the study's limitations, and how these limitations could be addressed in future research.

Additional comments

1. The quality of the figures needs to be improved. High-quality figures are essential for conveying the study's findings effectively, and improving their quality will enhance the manuscript's overall visual impact. Also, the Figure should include enough information, some figure in the manuscripts is too simple and unnecessary.
2. The manuscript should be improved by improving the English.

---

## Round 0.2 · accepted · Accept

Reviewers are satisfied with authors' revisions. I would suggest accepting this manuscript.

Reviewer 1 ·

Basic reporting

The author has responded to my comments and I have no further concerns

Experimental design

The author has responded to my comments and I have no further concerns

Validity of the findings

The author has responded to my comments and I have no further concerns

Reviewer 2 ·

Basic reporting

My concerns have been satisfactorily addressed and this work is now suitable for publication in PeerJ Computer Science.

Experimental design

no comment

Validity of the findings

no comment

Additional comments

no comment

Reviewer 4 ·

Basic reporting

Clear English used throughout

Experimental design

Experimental design is great

Validity of the findings

Findings is valifitied

Additional comments

NO